# Limited HIV-1 Subtype C *nef* 3′PPT Variation in Combination Antiretroviral Therapy Naïve and Experienced People Living with HIV in Botswana

**DOI:** 10.3390/pathogens10081027

**Published:** 2021-08-13

**Authors:** Kaelo K. Seatla, Dorcas Maruapula, Wonderful T. Choga, Olorato Morerinyane, Shahin Lockman, Vladimir Novitsky, Ishmael Kasvosve, Sikhulile Moyo, Simani Gaseitsiwe

**Affiliations:** 1Botswana Harvard AIDS Institute Partnership, Gaborone, Botswana; dmaruapula@bhp.org.bw (D.M.); wchoga@bhp.org.bw (W.T.C.); omorerinyane@bhp.org.bw (O.M.); slockman@hsph.harvard.edu (S.L.); smoyo@bhp.org.bw (S.M.); sgaseitsiwe@bhp.org.bw (S.G.); 2Faculty of Health Sciences, School of Allied Health Professions, University of Botswana, Gaborone, Botswana; kasvosvei@ub.ac.bw; 3Division of Human Genetics, Department of Pathology, Faculty of Health Sciences, University of Cape Town, Cape Town 7925, South Africa; 4Department of Immunology & Infectious Diseases, Harvard T.H. Chan School of Public Health, Boston, MA 02115, USA; 5The Warren Alpert Medical School of Brown University, Providence, RI 12321, USA; vnovi@hsph.harvard.edu; 6Division of Infectious Diseases, The Miriam Hospital, Providence, RI 23324, USA

**Keywords:** HIV-1, *nef*, Botswana, drug resistance mutations, 3′-polypurine tract, dolutegravir

## Abstract

Dolutegravir (DTG) is a potent anti-HIV drug that is used to treat HIV globally. There have been reports of mutations in the HIV-1 3′-polypurine tract (3′PPT) of the *nef* gene, contributing to DTG failure; however, there are limited ‘real-world’ data on this. In addition, there is a knowledge gap on the variability of 3′PPT residues in patients receiving combination antiretroviral therapy (cART) with and without viral load (VL) suppression. HIV-1 subtype C (HIV-1C) whole-genome sequences from cART naïve and experienced individuals were generated using next-generation sequencing. The *nef* gene sequences were trimmed from the generated whole-genome sequences using standard bioinformatics tools. In addition, we generated separate integrase and *nef* gene sequences by Sanger sequencing of plasma samples from individuals with virologic failure (VF) while on a DTG/raltegravir (RAL)-based cART. Analysis of 3′PPT residues was performed, and comparison of proportions computed using Pearson’s chi-square test with *p*-values < 0.05 was considered statistically significant. A total of 6009 HIV-1C full genome sequences were generated and had a median log_10_ HIV-1 VL (Q1, Q3) copies/mL of 1.60 (1.60, 2.60). A total of 12 matching integrase and *nef* gene sequences from therapy-experienced participants failing DTG/ RAL-based cART were generated. HIV-1C 3′PPT *nef* gene sequences from therapy-experienced patients failing DTG cART (*n* = 12), cART naïve individuals (*n* = 1263), and individuals on cART with and without virological suppression (*n* = 4696) all had a highly conserved 3′PPT motif with no statistically significant differences identified. Our study confirms the high conservation of the HIV-1 *nef* gene 3′PPT motif in ‘real-world’ patients and showed no differences in the motif according to VL suppression or INSTI-based cART failure. Future studies should explore other HIV-1 regions outside of the pol gene for associations with DTG failure.

## 1. Introduction

Dolutegravir (DTG) is a widely used second-generation integrase strand transfer inhibitor (INSTI) with a high genetic barrier to resistance [1,2,3,4,5]. It prevents the HIV integrase enzyme from incorporating viral DNA into the host cell genome [1]. However, resistance mutations within the integrase enzyme can cause reduced susceptibility to DTG [1,6]. Most major DTG resistance mutations amongst therapy-experienced individuals are usually located within the HIV-1 catalytic core domain of the integrase region [1,7]. However, a handful of studies have suggested associations between DTG failure and mutations outside the integrase region in the 3′-polypurine tract (3′PPT) of the HIV-1 *nef* gene [8,9].

The *nef* gene of HIV-1 is a small accessory protein of about 206 amino acids that contributes to HIV disease progression mainly by downregulating the expression of CD4 and major histocompatibility complex class I molecules, amongst other functions [10,11,12,13]. It has a highly conserved purine-dominated 15-nucleotide sequence (3′PPT) that is involved in the reverse transcription process, resulting in the production of double-stranded viral DNA, enabling the integration into the host cell genome [14,15,16].

Malet et al. unexpectedly cultured a virus that had mutations in the 3′PPT motif conferring resistance to DTG [8]. Similar mutations were identified in the guanine-tract (G-tract) motif at the 3′ end of 3′PPT of one patient with virologic failure (VF) while on a DTG monotherapy trial [9].

A subsequent study by Malet et al. with a larger number of individuals failing INSTI-based regimens revealed a highly conserved 3′PPT with no associations with DTG failure discernible [17]. Further in vitro work went on to confirm this [18]. A recent study from Cameroon amongst INSTI-naïve individuals also showed a highly conserved 3′PPT region [19]. Furthermore, analysis of publicly available HIV-1 *nef* gene sequences from the Los Alamos HIV-1 database reveals a highly conserved 3′PPT region across various subtypes [17].

Given these inconsistent study results and the fact that there are limited ‘real-world’ data on the contribution of 3′PPT to failure of DTG-based regimens, we conducted this study to address these knowledge gaps. In our study, we sought to determine the diversity of 3′PPT of the HIV-1 subtype C (HIV-1C) *nef* gene amongst cART-naïve and cART-treated individuals with and without VF. We also assessed whether HIV-1C mutations in 3′PPT contribute to VF amongst individuals failing INSTI-based cART regimens regardless of the presence of mutations in the integrase region.

## 2. Materials and Methods

### 2.1. Selection of Study Population and HIV-1 Genotyping

Participant samples were obtained from two studies conducted in Botswana. The first study consisted of sequences generated from residual plasma specimens obtained from therapy-experienced individuals experiencing VF while on DTG- or raltegravir (RAL)-based cART described elsewhere (BOSELE study; Figure 1) [7]. VF was defined as two or more consecutive plasma HIV-1 RNA levels (viral loads (VL)) > 400 copies/mL as per standard of care guidelines in Botswana. The HIV-1 integrase region was amplified using nested reverse transcription-polymerase chain reactions (RT-PCRs) where necessary and sequenced using a BigDye™ Terminator v3.1 Cycle Sequencing Kit (Applied Biosystems, Carlsbad, CA, USA) on a 3130xl Genetic Analyser (Life Technologies Corporation, Applied Biosystems, Carlsbad, CA, USA) as previously described [7,20]. Sequencing of the *nef* gene was attempted from the same HIV-1 extracts that integrase sequences were successfully generated from. Briefly, products of about 620 base pairs were amplified using nested RT-PCRs where necessary using the following primers numbered relative to HIV-1 reference strain (HxB2) nucleotide positions (shown in brackets): NEF8683F_pan TAGCAGTAGCTGRGKGRACAGATAG (8683–8707), NEF9536R_pan TACAGGCAAAAAGCAGCTGCTTATATGYAG (9507–9536), NEF8746_SgrI_AscI_F AGAGCACCGGCGCGCCTCCACATACCTASAAGAATMAGACARG (8736–8772), and NEF9474_SacII_ClaI_R GCCTCCGCGGATCGATCAGGCCACRCCTCCCTGGAAASKCCC (9449–9491) [21]. Amplicons were bidirectionally Sanger-sequenced as described above. The Sequencher software, version 5.0 (Gene Codes Corporation, Ann Arbor, MI, USA), was used to manually edit our electropherograms and form contigs with further downstream analysis carried using the BioEdit software.

The second group consisted of full-genome HIV-1C sequences obtained from participants enrolled in a large community-randomised HIV-1 prevention trial described elsewhere (BCPP study) [22,23]. The sequences were aligned to the HIV reference strain (HxB2) at the nucleotide level using virulign [24] and block-trimmed to the *nef* gene of HxB2 in the BioEdit, version 7.2.0, software [25]. We included two nucleotides before and one nucleotide after the 3′PPT *nef* gene sequence to have a complete amino acid coding for the 3′PPT tails. The sequences were assessed for hypermutations using the Hypermut tool at the Los Alamos National Laboratory HIV Database website (http://www.hiv.lanl.gov/ accessed on 8 March 2021). All sequences were exported to Microsoft^®^ Excel^®^ for Microsoft 365 MSO (16.0.13901.20148) 32-bit for further downstream analysis, and graphs were created. Additional statistical computations were performed using Stata version 14 (Stata Corporation, College Station, TX, USA) and R version 4.0.3; Pearson’s chi-square test was used to compare the proportion of 3′PPT of *nef* gene mutations per position by ART status and VL suppression. *p*-Values < 0.05 were considered statistically significant.

The Abbott RealTime HIV-1 assay, Cobas TaqMan/Cobas AmpliPrep HIV test (Roche Molecular Systems, Branchburg, NJ, USA), or Aptima HIV-1 Quant assay on Panther Systems (Hologic Inc., San Diego, CA, USA) was used to quantify HIV-1 RNA levels.

### 2.2. Ethical Statement

Both study protocols were approved by the health research and development division of the Botswana Ministry of Health and Wellness (Botswana’s IRB of authority). For the BOSELE study participants, a waiver of informed consent was obtained, and for the BCPP study, all the participants provided informed consent. The BCPP study was approved by the IRB at the U.S. Centers for Disease Control and Prevention and is registered at ClinicalTrials.gov (NCT01965470). All studies were conducted according to the principles stated in the Declaration of Helsinki.

## 3. Results

A total of 6021 HIV-1C *nef* gene sequences were available for analysis from both studies, and their basic demographics are shown in Table 1.

We included two nucleotides before (HxB2 nct position 9067 and 9068) and one nucleotide after 3′PPT (HxB2 nct position 9084) in our analysis of the 15-nucleotide 3′PPT region (5′ AAAAGAAAAGGGGGG 3′-HXB2 numbering 9069 to 9083) to complete the amino acid translation of the flanking regions of 3′PPT (Table 2, Figure 2).

All nucleotide positions of our HIV-1C *nef* gene 3′PPT sequences were highly conserved regardless of whether cART-naïve (*n* = 1263), on ART with VL < 400 (*n* = 4483) copies/mL, or on cART with VL ≥ 400 copies/mL (*n* = 213) groups (Figure 2). In addition, there was no statistically significant difference between ‘cART naïve’ and ‘on cART’ groups (*p* = 0.81), ‘on cART < 400′ and ‘on cART ≥ 400′ groups (*p* = 0.88), ‘ART naïve’ and ‘on cART < 400′ groups (*p* = 0.72), ‘cART naïve’ and ‘on cART ≥ 400′ groups (*p* = 0.99), ‘on cART and cART < 400′ groups (*p* = 0.86), ‘on cART and cART ≥ 400′ groups (*p* = 0.92), and ‘on ART’ and individuals with VF while on DTG/RAL cART group (*p* = 0.81). Analysis of sequences derived from buffy coat (*n* = 6009) adjusted for hypermutations (*n* = 2992) also revealed highly conserved *nef* 3′PPT residues with no statistically significant difference determined. The terminal six guanine stretch of 3′PPT also showed a high degree of conservation with all nucleotide residues having a mean rate of 99.47% (Figure 2, Appendix A).

## 4. Discussion

We analysed 12 HIV-1C *nef* 3′PPT sequences (eight had paired integrase sequences without INSTI drug resistance mutations and four had paired IN sequences with INSTI drug resistance mutations) from patients with VF while on DTG/RAL-based cART to search for changes in 3′PPT sequence that could be linked to DTG VF as previously reported, and we did not find any. Amongst the eight patients who were failing an INSTI-based regimen but who did not have INSTI resistance mutations in the integrase region, all had a 100% conservation in their 3′PPT sequences (they had no mutations at the nucleotide or amino acid level). In addition, we analysed 1263 3′PPT sequences from patients who were cART naïve to investigate HIV-1C 3′PPT variability. We further analysed 4696 3′PPT sequences from patients on cART (but not on a DTG-based regimen) stratified according to virological suppression and found no 3′PPT region variability.

Malet et al. found some significant variability at position 9071 (25% and 10% in HIV-1 subtype B and CRF01, respectively) [17] and position 9075 (10% variability in HIV-1 subtype D). In our analysis, position 9071 revealed a variability of 0.29% (*n* = 6009) and 0% (*n* = 12) of sequences from patients not exposed to DTG cART and those failing DTG/RAL cART, respectively. Position 9075 was also conserved with a variability of 0.4% (*n* = 6009) and 0% (*n* = 12) amongst the two groups. Perhaps this difference in variability could be explained by the different HIV-1 subtypes—all our sequences were HIV-1C.

We observed a high conservation amongst the six nucleotides of the G-tract residues of 3′PPT (mean of 99.47%) similar to what others have found (99.95%) [8,17,18,19].

We did not explore the entire HIV-1C genome (5′ long terminal repeat (LTR), gag, protease, reverse transcriptase, vif, vpr, vpu, envelope, and 3′ LTR) for other mutations that could be linked to INSTI resistance. We did not measure plasma DTG or RAL levels to check whether issues of nonadherence could be contributing to VF.

In conclusion, we conducted one of the largest analyses of the HIV-1C 3′PPT region, showing great conservation of the region at the nucleotide and amino acid level. Although we did not detect any association of 3′PPT mutations with VF on INSTI-based cART, our data were limited on the number of 3′PPT sequences generated from patients failing an INSTI-containing regimen without INSTI mutations. However, our data add to a growing list of studies that have found no association of 3′PPT mutations with INSTI resistance [17,18]. Future studies should also investigate the role of other HIV-1 genes outside of Pol as this might enhance our understanding of mechanisms associated with INSTI resistance.

## Figures and Tables

**Figure 1 pathogens-10-01027-f001:**
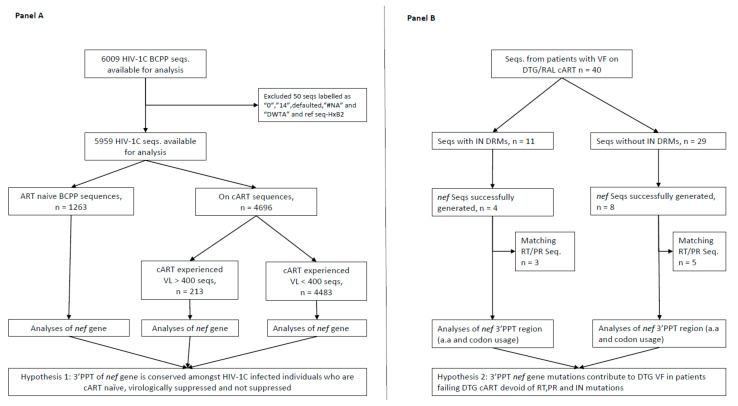
Study schema depicting the selection of study sequences. (**A**), selection and analysis of 3′PPT *nef* gene sequences according to participant cART status and HIV-1 RNA levels; (**B**), selection and analysis of 3′PPT of *nef* gene amongst individuals with VF while on DTG/RAL-based cART. Seqs, sequences; PID, participant identification number; VL, viral load; VF, virologic failure; RAL, raltegravir; DTG, dolutegravir; cART, combination antiretroviral therapy; 3′PPT, 3′-polypurine tract; RT, reverse transcriptase HIV-1 gene; PR, protease HIV-1 gene; DRMs, drug resistance mutations; HIV-1C, HIV-1 subtype C.

**Figure 2 pathogens-10-01027-f002:**
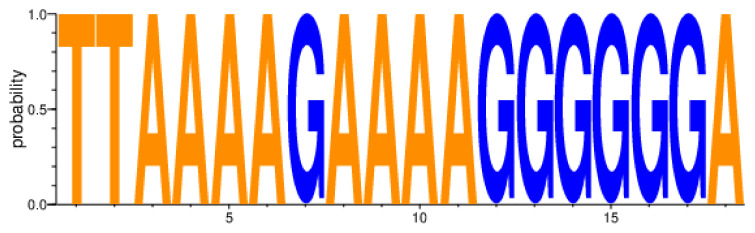
HIV-1C *nef* gene 3′PPT variability amongst 6009 sequences from individuals on cART and not on cART. 3′PPT, 3′polypurine tract; cART, combination antiretroviral therapy; HIV-1C, HIV-1 subtype C.

**Table 1 pathogens-10-01027-t001:** Basic demographics of 6009 HIV-1C diagnosed, cART-naïve, and cART-experienced individuals.

Basic Characteristics		* HIV-1C Diagnosed Participant Sequences Available for Analysis (*n* = 6009)	** Sequences from Participants with VF on DTG cART (*n* = 40)
**Age (years), median (Q1, Q3)**		40 (33, 48)	41 (26, 45)
^†^ Gender	Female n (%)	4241 (71%)	15 (44%)
	Male n (%)	1757 (29%)	19 (56%)
	Unknown n (%)	11 (0.2%)	** N/A
Median log _10_ HIV-1 RNA (Q1, Q3) copies/mL		1.60 (1.60, 2.60)	4.53 (3.98, 5.10)

* Participants from BCPP study, ** We generated 40 sequences representing 34 ‘unique’ individuals from an ongoing study characterizing therapy-experienced participants experiencing VF while on DTG/RAL cART. ^†^ Analysable gender data available for 5998 individuals; rest of dataset contained ‘ND’ shown as ‘Unknown’. ND, not documented; VF, virologic failure; DTG, dolutegravir; cART, combination antiretroviral therapy; BCPP, Botswana Combination Prevention Project. Column 3 of this table is the same dataset represented in Column 3 of Table 1 of Seatla et al. [7]. This table has been modified with permission from Seatla et al. [7].

**Table 2 pathogens-10-01027-t002:** HIV-1C *nef* 3′PPT variability amongst 12 therapy-experienced individuals experiencing VF while on DTG/RAL cART.

	^Β^ Major DRMs					3′PPT of the HIV-1 *nef* Gene															
HxB2_nct_positions				9067	9068	9069	9070	9071	9072	9073	9074	9075	9076	9077	9078	9079	9080	9081	9082	9083	9084
HxB2_NEF_nct position				271	272	273	274	275	276	277	278	279	280	281	282	283	284	285	286	287	288
HXB2_NEF_gene_seqs.	RT (NRTI; NNRTI)	PI	INSTI	T	T	A	A	A	A	G	A	A	A	A	G	G	G	G	G	G	A
^¥^ 139-0001-8	M41L, D67N, K70KR, V75M, M184V, L210W, T215Y, K219E; A98G, Y181C, G190A	M46I, I47V, I54L, L76V, I84V, Q58E, N83D	E138K, S147G, Q148R, N155H, (E157Q)	T	T	A	A	A	A	G	A	A	A	A	G	G	G	G	G	G	A
^¥^ 139-0002-8	^±^ K70R, M184V;K219N/Y181C (20APRIL2009) ^±^ D67N, K70R, M184V/NONE (18AUG2016)	^±^ V32I, I47V, I54L, I84V (20 APRIL2009) ^±^ V32I, I47V, I54L, I84V (18AUG2016)	E138K, G140A, S147G, Q148R, (T97A)	T	T	A	A	A	A	G	A	A	A	A	G	G	G	G	G	G	A
^¥^ 139-0004-6	^±^ M41L, T69G, K70R, M184V, T215C, K219E; A98G, K101E	^±^ M46I, I54V, L76V, V82A	T66A, G118R, E138EAKT	T	T	A	A	A	A	G	A	A	A	A	G	G	G	G	G	G	A
139-0005-3	* M184V; A98G	* Q58E	* ND	T	T	A	A	A	A	G	A	A	A	A	G	G	G	G	G	G	A
^¥^ 139-0012-9	^±^ M184V, M41L, T215Y; ND (9 June 2010)	^±^ M46I,V82A Accessory; L10F, L24I (9 June 2010)	N155NH (D232DN)	T	T	A	A	A	A	G	A	A	A	A	G	G	G	G	G	G	A
139-0013-0	^±^ M184V, A62V, M41L (28 July 2011); ^±^ E138K (3 February 2016), ^±^ * E138K (24 January 2018)	^±^ V11IV (28 July 2011)	ND	T	T	A	A	A	A	G	A	A	A	A	G	G	G	G	G	G	A
139-0015-4	* ND	* ND	* ND	T	T	A	A	A	A	G	A	A	A	A	G	G	G	G	G	G	A
139-0017-2	* ND	* ND	* ND	T	T	A	A	A	A	G	A	A	A	A	G	G	G	G	G	G	A
139-0018-3	^±^ D67G, K70E, M184V; Y181C, G190A	* ND	* ND	T	T	A	A	A	A	G	A	A	A	A	G	G	G	G	G	G	A
139-0021-4	* K65N; V179D	* ND	* ND	T	T	A	A	A	A	G	A	A	A	A	G	G	G	G	G	G	A
139-0026-8	* ND	* ND	* ND	T	T	A	A	A	A	G	A	A	A	A	G	G	G	G	G	G	A
139-0119-5	* ND; K103N, P225H	* ND	* ND	T	T	A	A	A	A	G	A	A	A	A	G	G	G	G	G	G	A
				Leucine, Leu, L			Lysine, Lys, K			Glutamic acid, Glu, E			Lysine, Lys, K			Glycine, Gly, G			Glycine, Gly, G		

^Β^ Major DRMs assessed by using the Stanford HIV drug resistance database; * denotes that RT-PCR testing for IN, RT, and PR was performed on the same (unique) sample from each patient. ^±^ Historical DRMs denoted with ‘^±^’ retrieved from electronic databases and/or patients’ medical charts. Mutations listed within brackets ‘()’ are accessory mutations. ^¥^ denotes the same participants as listed in Table 2 of Seatla et al. [7]. ND, no mutations detected; cART, combination antiretroviral therapy; GRT, genotypic resistance testing; RT, reverse transcriptase; NRTI, nucleoside/nucleotide reverse transcriptase inhibitors; NNRTI, non-nucleoside reverse transcriptase inhibitors; PR, protease; PI, protease inhibitor; HxB2, HIV reference sequence_K03455; INSTI, integrase strand transfer inhibitors; DRMs, drug resistance mutations. Light blue colour depicts the 3′PPT of the HIV-1 *nef* gene, yellow and orange colours depict the amino acid translation of the 3-nucleotide sequence. Adapted from Figure 2a of Malet et al. [8], Figure 1 of Malet I et al. [17], and with permission from Table 2 of Seatla et al. [7].

## Data Availability

Sequences available at national centre for biotechnology information (NCBI) GenBank, accession numbers MW690052-MW690089, MG989443.1, MG989444.

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
