# Peer review of "Limited HIV-1 Subtype C *nef* 3′PPT Variation in Combination Antiretroviral Therapy Naïve and Experienced People Living with HIV in Botswana"

_pathogens, 2021, doi:10.3390/pathogens10081027_

Round 1

Reviewer 1 Report

The work by Seatla et al. analyses the presence of mutations in the nef-3'PPT region of two cohorts of HIV-1 subtype C patients, to assess their possible impact on either Dolutegravir resistance and virological failure.

I suggest the Authors to carefully check English and punctuation throughout the manuscript, because there are several minor typos. I also recommend to use abbreviations once they interoduce them in the text (e.g. Dolutegravir is abbreviated as DTG but then occurs again in the text in its extended form).

Also, be sure that the gene names are written in the correct form (italics, not capitalised) and do not change throughout the text (3'PPT vs 3'-PPT, etc)

It is not clear what do authors mean when in the abstract they state that "We generated HIV-1 subtype C (HIV-1C) using next-generation sequencing from cART naïve and experienced individuals". I think they meant they sequenced specific portions of HIV-1 C genome, with the primers indicated in Methods section. If this is correct, please rephrase this sentence because in the present form it is misleading.  

The introduction in my opinion is not sufficient to provide the readers with an adequate knowledge of the topic. Beside the fact that it gives very few data about the general state of the art, it also lack the minimum information to understand the study rationale. For example, why was subtype C specifically choose for this analysis? Is it associated with higher mutation rate in IN/nef/ppt that can be hence relevant to DTG resistance and/or higher virological failure? I also thought that they choose this specific subtype to compare the obtained results with previous studies using the same, but I don't think this is the case given that they state in the discussion that "Malet et al found some significant variability at position 9071 (25% and 10% in HIV-1 subtype B and CRF01 respectively [4] and position 9075 (10% variability in HIV-1 subtype D)" and even that the differences observed in their results perhaps "could be explained by the different HIV-1 subtypes- all our sequences were HIV-1C". 

The actual number of samples that were analysed to obtain the final results is also not clearly stated, and no reference to the two panel of figure 1 is provided in the methods text describing the two populations. For example, sequences from patients with virological failure are reported to be 40, but then it seems from figure 1 that only 4 + 8 new genes (12 in total) were successfully sequenced. I think that the authors should state clearly in the results the final amount of samples on which were actually based their observations.

Figure 2 should be instead Table 2. Concerning figure 3, what does the letter "B" on the top mean?

The meaning of "INSTI DRMs" should be explained also in the main text 

Author Response

Reviewer_1_round_1_Comments and Responses_22July2021

“The work by Seatla et al. analyses the presence of mutations in the nef-3'PPT region of two cohorts of HIV-1 subtype C patients, to assess their possible impact on either Dolutegravir resistance and virological failure.”

Point 1

“I suggest the Authors to carefully check English and punctuation throughout the manuscript, because there are several minor typos.”

Response to point 1

Thank you for pointing this out. English and punctuation have been checked throughout and corrected were necesary

Point 2

“I also recommend to use abbreviations once they interoduce them in the text (e.g. Dolutegravir is abbreviated as DTG but then occurs again in the text in its extended form).”

Response to point 2

Thank you for pointing this out.

Dolutegravir changed to DTG in line 39,63.

“virological failure” changed to “VF” in line 38, 133, 183, “virological failure” deleted in line 65 and left “VF”

“viral load” changed to “VL” in line 114. Deleted “viral load” in line 133 of table 1 legend

Point 3

“Also, be sure that the gene names are written in the correct form (italics, not capitalised) and do not change throughout the text (3'PPT vs 3'-PPT, etc)”

Response to point 3

Thank you for highlighting this important point.

We have adopted to using “3'PPT” instead of “3'-PPT” and this has been changed throughout the manuscript.

We have corrected instances written as “ NEF, Nef, nef ” to the correct form (italics, not capitalised) “ nef ” in lines 2,5,22,27,32,33,37,41,49,51,59,61,65,80,95,96,105,106,114,128,141,150,157, Figure 2 first row heading “ 3’PPT of the HIV-1 nef gene” and legend, Line 80, changed following sentence “Nef gene sequencing…” to “Sequencing of nef gene…”.

NB: The names of the RT-PCR and sequencing primers (line 84-88) were not changed i.e. NEF8683F_pan

Point 4

“It is not clear what do authors mean when in the abstract they state that "We generated HIV-1 subtype C (HIV-1C) using next-generation sequencing from cART naïve and experienced individuals". I think they meant they sequenced specific portions of HIV-1 C genome, with the primers indicated in Methods section. If this is correct, please rephrase this sentence because in the present form it is misleading.   “

Response to point 4

Thank you for highlighting this point and the suggestion.

We appreciate how that sentence can cause misinterpretations. Whole genome HIV-1C sequences were generated using NGS. The current analysis focused on the HIV-1 nef gene, so the nef gene sequences were trimmed from the whole genome sequences using standard bioinformatics tools. Detailed methods explained in line 101-114 and panel A of figure 1.

To make this clearer for the journal’s readership in the abstract, we have changed the sentence in abstract (line 25) “We generated HIV-1 subtype C (HIV-1C) using next-generation sequencing from cART naïve and experienced individuals.” to “HIV-1 subtype C (HIV-1C) whole genome sequences from cART naïve and experienced individuals were generated using next-generation sequencing. The nef gene sequences were trimmed from the generated whole genome sequences using standard bioinformatics tools.”

In addition to the abstract, we have removed the following sub-headings “BACKGROUND” (line 21), “METHODS” (line 25), “RESULTS” (line 30) and “CONCLUSION” (line 36).

Point 5a

“The introduction in my opinion is not sufficient to provide the readers with an adequate knowledge of the topic. Beside the fact that it gives very few data about the general state of the art, it also lack the minimum information to understand the study rationale. For example, why was subtype C specifically choose for this analysis? Is it associated with higher mutation rate in IN/nef/ppt that can be hence relevant to DTG resistance and/or higher virological failure?”

Response to point 5a

Thank you for raising these important concerns.

We have added more information on DTG and references (line 47-51).

We have further ‘streamlined’ transition into the 3’PPT topic by rearranging some words and English grammar (line 53-54).

We have gone-on to re-search databases for new data on 3’PPT and DTG; there is very limited real-world data on DTG failures and 3’PPT sequence variations from HIV-1C infected patients besides the already mentioned few data in the manuscript (line 53-70).

We go on to express the global knowledge gap on possible impact of 3’PPT sequence variation and differences in viral load status (VL below or greater than 400 cps/mL) in real-world HIV-1C infected patients on cART (line 66-69). To the best of our knowledge, we might be amongst the first groups to report on this exploration albeit the negative findings.

We did not specifically select to conduct this study on subtype C. HIV-1C is the predominant circulating variant in our setting (Botswana).

Investigating the association between DTG failure and sequence variations in IN/nef/ppt amongst different HIV-1 subtypes was not the focus of this manuscript.

We appreciate the interesting questions, and it is something worth exploring in future.

Point 5b

“I also thought that they choose this specific subtype to compare the obtained results with previous studies using the same, but I don't think this is the case given that they state in the discussion that "Malet et al found some significant variability at position 9071 (25% and 10% in HIV-1 subtype B and CRF01 respectively [4] and position 9075 (10% variability in HIV-1 subtype D)" and even that the differences observed in their results perhaps "could be explained by the different HIV-1 subtypes- all our sequences were HIV-1C".  “

Response to point 5b

Thank you for highlighting this point.

We did not specifically select to conduct this study on subtype C. HIV-1C is the predominant circulating variant in our setting (Botswana).

Investigating the association between DTG failure and sequence variations in IN/nef/ppt amongst different HIV-1 subtypes was not the focus of this manuscript. We appreciate the interesting questions, and it is something worth exploring in future.

Point 6

“The actual number of samples that were analysed to obtain the final results is also not clearly stated, and no reference to the two panel of figure 1 is provided in the methods text describing the two populations. For example, sequences from patients with virological failure are reported to be 40, but then it seems from figure 1 that only 4 + 8 new genes (12 in total) were successfully sequenced. I think that the authors should state clearly in the results the final amount of samples on which were actually based their observations.”

Response to point 6

Thank you for highlighting this important point.

The two panel figure 1 was referenced in the materials and methods section line 75 “ the first study consisted of sequences generated from residual plasma specimens obtained from therapy-experienced individuals experiencing VF while on DTG- or raltegravir (RAL)-based cART (BOSELE study, described elsewhere) Figure 1 [1].”

Point 7

“Figure 2 should be instead Table 2.”

Response to point 1

Thank you for pointing this out. This has been corrected

Point 8

“ Concerning figure 3, what does the letter "B" on the top mean?”

Response to point 8

Thank you for pointing this out. The “B” has been deleted

Point 9

“The meaning of "INSTI DRMs" should be explained also in the main text”

Response to point 9

Thank you for pointing this out. DRM has been written in full in body of text line 18-19 “ We analysed 12 HIV-1C nef 3’PPT sequences (8 had paired integrase sequences without INSTI drug resistance mutations and 4 had paired IN sequences with INSTI drug resistance mutations)…”

Reviewer 2 Report

In this article, Seatla and colleagues have compared the 3'-PPT variation in HIV-1C amongst individuals with and without virological failure. Although there have been a couple of other papers that have suggested 3'-PPT mutations might confer anti-ART immunity to HIV, however, these findings were either incidental (in small data sets) or generated through in vitro directed evolution of the virus. The new data set (of genome sequences) generated for this study is considerably small and therefore the data presented in this study is somewhat an incremental and descriptive advance. The conclusions drawn need to be reconsidered and a thorough discussion of the limitations of the study needs to be included in the article. It would have been of interest to readers to include more extensive sequence analyses (e.g. focusing on other genes such as IN) in combination with currently reported results would have made this project more robust. 

Author Response

Reviewer_2_round_1_Comments and Responses_22July2021

“In this article, Seatla and colleagues have compared the 3'-PPT variation in HIV-1C amongst individuals with and without virological failure.”

 “Although there have been a couple of other papers that have suggested 3'-PPT mutations might confer anti-ART immunity to HIV, however, these findings were either incidental (in small data sets) or generated through in vitro directed evolution of the virus.”

Point 1

“The new data set (of genome sequences) generated for this study is considerably small and therefore the data presented in this study is somewhat an incremental and descriptive advance.”

Response to point 1

Thank you for highlighting this point. We had already alluded to our small sample size as a limitation in the discussion section line 43-46 “Although we did not detect any association of HIV-1 3’PPT mutations with VF on INSTI based cART, our data was limited on the number of HIV-1 3’PPT sequences generated from patients failing INSTI containing regimen without INSTI mutations.” However, our data provides the largest set of contemporaneous HIV-1C 3’PPT sequences that are informative on baseline 3’PPT variation. This will be an important comparator group for future studies generating more 3’PPT sequences from patients failing INSTI containing regimen.

Point 2

“The conclusions drawn need to be reconsidered and a thorough discussion of the limitations of the study needs to be included in the article.”

Response to point 2

Thank you for this suggestion.

We have alluded to the limitations of the dataset but still this work is important as it;

  1. Provides a large sequence dataset of HIV-1C 3’PPT sequences from DTG naïve patients that will be useful as comparator group once more HIV-1C 3’PPT sequences are generated from patients failing DTG containing cART.
  2. Provides the first HIV-1C 3’PPT sequences from patients failing DTG based cART in Botswana and finds no indication of mutations associated with DTG resistance in previous studies.

Although there were limited sequences from patients failing DTG, future projects will build on this results.

Point 3

It would have been of interest to readers to include more extensive sequence analyses (e.g. focusing on other genes such as IN) in combination with currently reported results would have made this project more robust.  “

Response to point 3

Thank you for pointing this out.

Integrase sequences were analysed, and the current project differentiates those with INSTI resistance mutations and those that do not harbour INSTI resistance mutations. In some instances, RT and PI sequences were also generated and analysed for DRMs.

Round 2

Reviewer 2 Report

The authors have addressed all my comments. 

Author Response

Thank you for the suggestions. The introduction has been improved with more background and references added; line 56 to 60 and 72 to 76. The conclusion has also been improved; line 211 and 218